# Association between hospital liver transplantation volume and mortality after liver re-transplantation

**Seung-Young Oh[1,2], Eun Jin Jang[3], Ga Hee Kim[4], Hannah Lee[1,5], Nam-Joon Yi[2], Seokha Yoo[5], Bo Rim Kim[5], Ho Geol Ryu[1,5]***

**1** Critical Care Center, Seoul National University College of Medicine, Seoul National University Hospital, Seoul, Korea, **2** Department of Surgery, Seoul National University College of Medicine, Seoul National University Hospital, Seoul, Korea, **3** Department of Information Statistics, Andong National University, Gyeongsangbuk-do, Korea, **4** Department of Statistics, Kyungpook National University, Daegu, Korea, **5** Department of Anesthesiology, Seoul National University College of Medicine, Seoul National University Hospital, Seoul, Korea

* hogeol@gmail.com

**Data Availability Statement:** The data used in this study are third party data from the National Health Insurance Service (https://nhiss.nhis.or.kr/bd/ay/bdaya001iv.do) and can be accessed following the protocol outlined in the Methods section.

## Abstract

### Background

The relationship between institutional liver transplantation (LT) case volume and clinical outcomes after liver re-transplantation is yet to be determined.

### Methods

Patients who underwent liver re-transplantation between 2007 and 2016 were selected from the Korean National Healthcare Insurance Service database. Liver transplant centers were categorized to either high-volume centers ($\geq$ 64 LTs/year) or low-volume centers (< 64 LTs/year) according to the annual LT case volume. In-hospital and long-term mortality after liver re-transplantation were compared.

### Results

A total of 258 liver re-transplantations were performed during the study period: 175 liver re-transplantations were performed in 3 high-volume centers and 83 were performed in 21 low-volume centers. In-hospital mortality after liver re-transplantation in high and low-volume centers were 25% and 36% ($P = 0.069$), respectively. Adjusted in-hospital mortality was not different between low and high-volume centers. Adjusted 1-year mortality was significantly higher in low-volume centers (OR 2.14, 95% CI 1.05–4.37, $P = 0.037$) compared to high-volume centers. Long-term survival for up to 9 years was also superior in high-volume centers ($P = 0.005$). Other risk factors of in-hospital mortality and 1-year mortality included female sex and higher Elixhauser comorbidity index.

**Funding:** The author(s) received no specific funding for this work.

**Competing interests:** The authors have declared that no competing interests exist.

## Conclusion

Centers with higher case volume ($\geq$ 64 LTs/year) showed lower in-hospital and overall mortality after liver re-transplantation compared to low-volume centers.

## Introduction

Liver re-transplantation is the only remaining option for survival in patients who develop graft failure after their primary liver transplantation (LT) [1, 2] and the number of liver re-transplantations are increasing in proportion to the number of primary LTs being performed [3]. Major indications for liver re-transplantation include primary non-function, vascular thrombosis, disease recurrence, graft rejection, and biliary complication [1, 4–7]. The reported proportion of liver re-transplantations among LTs range between 10% and 17% [4, 5, 8].

Poor post-transplant survival after liver re-transplantation compared to primary LT have been attributed to surgical complexity and disease progression during the wait time [5, 9, 10] Identified risk factors of poor outcome after liver re-transplantation include higher Model for end-stage liver disease (MELD) scores, old recipient age, cause of graft failure, and prolonged interval between primary LT and liver re-transplantation [5, 7, 11, 12].

Institutional case volume has been known to be associated with improved outcomes after high risk surgery such as coronary artery bypass, pancreatectomy, and esophagectomy [13–16]. Considering that liver re-transplantation is one of the most technically challenging surgical procedures, the impact of case volume may be most prominent. However, in contrast to living or deceased donor LT, data supporting case volume effect of liver re-transplantation are lacking [17].

The aim of this study was to evaluate the case volume effect on short and long-term outcomes after liver re-transplantation.

## Material and methods

The study protocol of this retrospective cohort study was approved by the institutional review board of Seoul National University Hospital (No. 1704-004-840).

### Data source and study population

The National Healthcare Insurance Service (NHIS) database contains all claims data of the Korean population covered under the National Healthcare Insurance (NHI) program and the Medical Aid program in Korea. The NHIS database is provided after de-identification for research purposes [18].

We identified adult patients (age $\geq$ 18) who received liver re-transplantation between 2007 and 2016 from the NHIS database by searching NHI procedure codes for liver re-transplantation with living donor (Q8145 –Q8450) and liver re-transplantation with deceased donor (Q8140 –Q8144) during hospitalization. After identification of adult liver re-transplantation recipients, underlying comorbidities such as hypertension, diabetes mellitus, coronary artery disease, and chronic kidney disease, and cardiovascular disease were extracted from the database using ICD-10 codes. The Elixhauser comorbidity index, derived from 30 disease entities using ICD-10 codes [19], was incorporated to adjust for severity of illness. The Elixhauser comorbidity index has been shown to correlate with hospital mortality [20] and is frequently used in health service research to adjust for confounders or to represent patient population

characteristics. A recent study had suggested superiority of the Elixhauser comorbidity system compared to the previously used Charlson comorbidity system at adjusting for comorbidity [21]. Coexisting liver disease such as hepatitis A virus, hepatitis B virus, hepatitis C virus, hepatocellular carcinoma, alcoholic cirrhosis, and primary biliary sclerosis were also extracted using ICD-10 codes. Since there was no accurate date information for the primary LT and re-transplantation, re-transplantations were classified into early re-transplantation (both primary LT and re-transplantation in the same hospital admission) and late re-transplantation (re-transplantation only).

Outcomes such as in-hospital mortality, intensive care unit (ICU) length of stay, and hospital length of stay were also extracted. Long-term mortality was determined when death was reported to the NHI for termination of healthcare coverage by the NHI.

### Definition of case volume

The case volume of each institution was defined as the average annual number of LTs, including living donor liver transplantation (LDLT), deceased donor liver transplantation (DDLT), and re-transplantation. To determine a cut-off for dividing low and high-volume centers, receiver operating characteristic (ROC) curve analysis between institutional case volume and in-hospital mortality was performed. Area under the curve (AUC) was 0.557 and the optimal cut-off that made maximum Youden-index was 63.21 LTs/year (Fig 1). According to this result, centers were categorized to either low-volume centers (< 64 LTs/year) or high-volume centers (≥ 64 LTs/year) depending on the case volume.

### Organ allocation policy for liver re-transplantation in Korea

Patients who develop primary non-function or hepatic artery thrombosis within 1 week after primary LT and require liver re-transplantation are granted priority status. Patients with priority status can stay on the waiting list as a highly urgent candidate for up to 1 week which can be extended for on additional week if there are no donors. Other candidates for liver re-transplantation are entitled to the same status as patients waiting for primary LT.

### Statistical analysis

Patient characteristics were compared according to case volume using the independent t-test for continuous variables and chi-square test or Fisher's exact test for categorical variables, respectively. After adjusting for age, sex, transplantation period, and Elixhauser comorbidity index, the in-hospital mortality after liver re-transplantation was assessed according to case volume using logistic regression. The goodness-of fit for the logistic regression model was assessed using Hosmer-Lemeshow test. Survival after liver re-transplantation according to case volume were compared using Cox proportional hazard model after adjusting for age, sex, and Elixhauser comorbidity index. The Kaplan-Meier survival analysis after liver re-transplantation according to case volume and log-rank test to compare the survival curve was performed. The goodness-of fit for the Cox proportional hazard model was assessed using the likelihood ratio test and the proportional hazard assumption was explored using the log-minus-log plot. Mean and standard deviation (SD) for liver re-transplantation outcomes (ICU length of stay and hospital length of stay) according to case volume were presented and compared using the independent t-test.

All analyses were performed using SAS 9.4 (SAS Institute, Cary, NC). Results were considered statistically significant when $P$-values were less than 0.05.

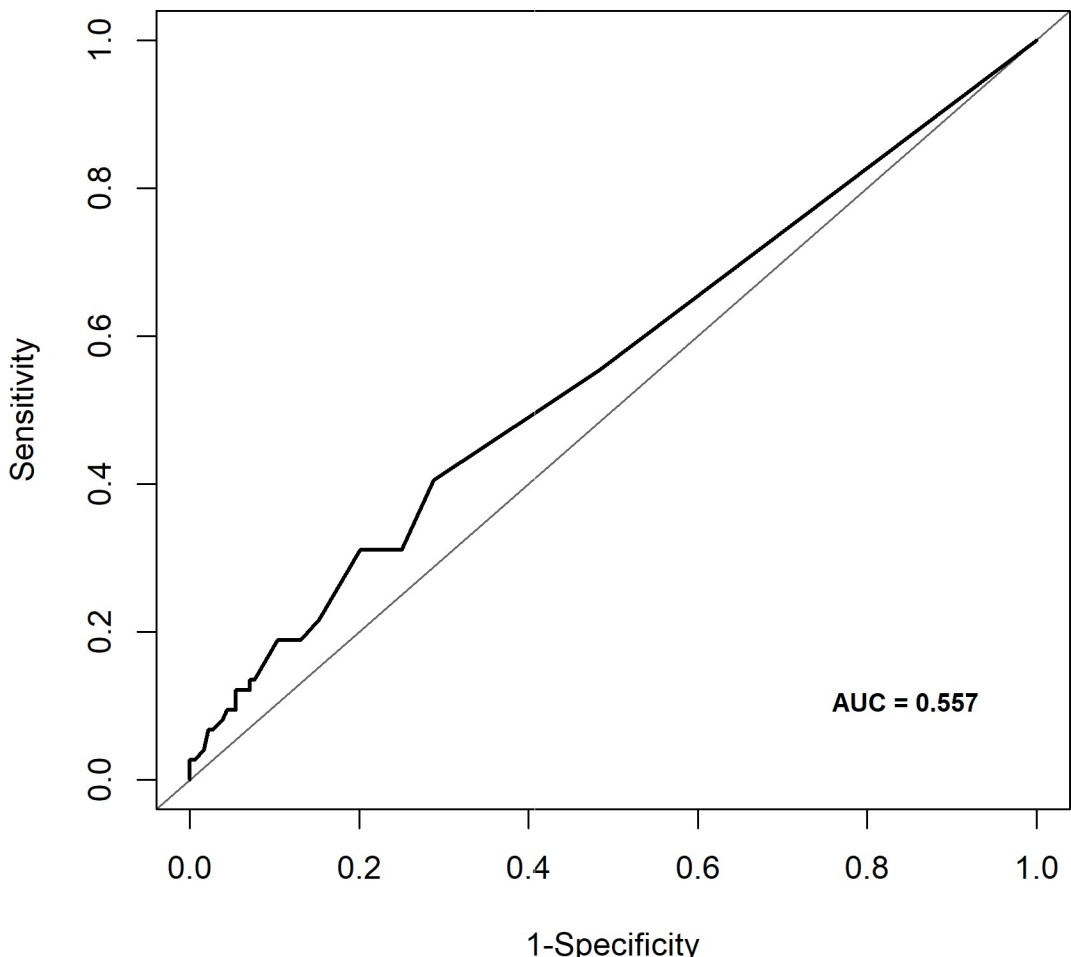

**Fig 1. Receiver operating characteristic (ROC) curve analysis to determine a cut-off for dividing low and high-volume centers.**

## Results

A total 258 liver re-transplantations were performed in 24 centers from January 2007 to December 2016 in Korea. Three high-volume centers performed 175 (67.8%) liver re-transplantations, while 21 low-volume centers performed 83 (32.2%) liver re-transplantations (Table 1 and Fig 2). There was no significant difference in Elixhauser comorbidity index between patients in high-volume and low-volume centers (21.9 vs. 21.5, *P* = 0.715).

Among 24 centers that performed liver re-transplantation during the study period, there were 3 high-volume centers and 21 low-volume centers. There were three centers with a 100% in-hospital mortality rate after liver re-transplantation, and all three centers had two cases or less during the study period.

The in-hospital mortality rate after liver re-transplantation was 28.7% (74/258); 25.1% (44/175) in high-volume centers and 36.1% (30/83) in low-volume centers (*P* = 0.069). Although time periods were included in the multivariable analyses to adjust for temporal trends during the 10 year of study period [9, 22], there was no difference in in-hospital mortality between time periods. After adjustment, 60 years or older (OR 2.39, 95% CI [1.02, 5.60], *P* = 0.044), female (OR 2.05, 95% CI [1.14, 4.10], *P* = 0.032), liver re-transplantation in low-volume centers (OR 1.93, 95% CI [1.00, 3.73], *P* = 0.049), hepatitis A infection (OR 4.15, 95% CI

**Table 1. Patient characteristics.**

| | Low-volume center (< 64 LTs/year) | High-volume center (≥ 64 LTs/year) | *P*-value |
|---|---|---|---|
| Number of centers | 21 | 3 | |
| Annual number of LTs | 12.4 (1.3, 63.1) | 114.6 (95.5, 327.4) | < 0.001 |
| Annual number of liver re-transplantations | 0.7 (0.1, 1.8) | 3.0 (2.1, 12.8) | < 0.001 |
| Age | 54 (23, 68) | 52 (19,74) | 0.893 |
| Sex (M:F) | 56:27 | 111:64 | 0.526 |
| Comorbidities | | | |
| Hypertension | 53 (63.8) | 99 (56.5) | 0.267 |
| Diabetes mellitus | 25 (30.1) | 79 (45.1) | 0.022 |
| Coronary artery disease | 8 (9.6) | 17 (9.7) | 0.985 |
| Chronic kidney disease | 4 (4.8) | 24 (13.7) | 0.032 |
| Cerebrovascular disease | 2 (2.4) | 8 (4.6) | 0.508 |
| Elixhauser comorbidity index | 21.5 ± 8.2 | 21.9 ± 7.9 | 0.715 |
| Coexisting liver disease | | | |
| Hepatitis A virus | 4 (4.8) | 7 (4.0) | 0.750 |
| Hepatitis B virus | 54 (65.1) | 106 (60.6) | 0.488 |
| Hepatitis C virus | 10 (12.0) | 38 (21.7) | 0.062 |
| Hepatocellular carcinoma | 41 (49.4) | 66 (37.7) | 0.075 |
| Alcoholic liver cirrhosis | 13 (15.7) | 13 (13.1) | 0.585 |
| Primary biliary sclerosis | 2 (2.4) | 5 (2.9) | 1.000 |
| Timing of re-transplantation | | | < 0.001 |
| Early re-transplantation | 45 (54.2) | 51 (29.1) | |
| Late re-transplantation | 38 (45.8) | 124 (70.9) | |
| Donor type of re-transplantation | | | 0.549 |
| Living donor | 14 (16.9) | 35 (20.0) | |
| Deceased donor | 69 (83.1) | 140 (80.0) | |
| Total number of primary LTs | 4684 | 6027 | |
| Proportion of re-transplantation | | | |
| Early re-transplantation | 1.0 | 0.8 | |
| Late re-transplantation | 0.8 | 2.1 | |

Data are presented as median (range), number (%), mean ± SD, percentage.

LT, liver transplantation.

[1.09, 15.84], *P* = 0.037), early re-transplantation (OR 1.26, 95% CI [0.65, 2.45], P = 0.049), graft from deceased donor (OR 7.75, 95% CI [2.13, 28.23], P = 0.002), and liver re-transplantation between 2011 and 2013 (OR 2.11, 95% CI [1.03, 4.34], *P* = 0.042) were identified as risk factors of in-hospital mortality after liver re-transplantation (Table 2).

The overall 1-year mortality rate after liver re-transplantation was 36.8% (81/220); 32.2% (48/149) in high-volume centers and 46.5% (33/71) in low-volume centers (*P* = 0.041). Low-volume centers showed a significantly higher 1-year mortality compared to high-volume centers (OR 2.54, 95% CI [1.24, 5.21], *P* = 0.011) after adjusting for relevant factors. In addition to case volume, older age (≥ 60 years), presence of hepatitis C virus, graft from deceased donor, and liver re-transplantation before 2014 were identified as significant risk factors of 1-year mortality after liver re-transplantation (Table 3).

Evaluation of long-term survival for up to 9 years after liver re-transplantation showed lower survival in patients who received liver re-transplantation in low-volume centers compared to high-volume centers (*P* = 0.002) (Fig 3). Multivariable Cox regression analysis also

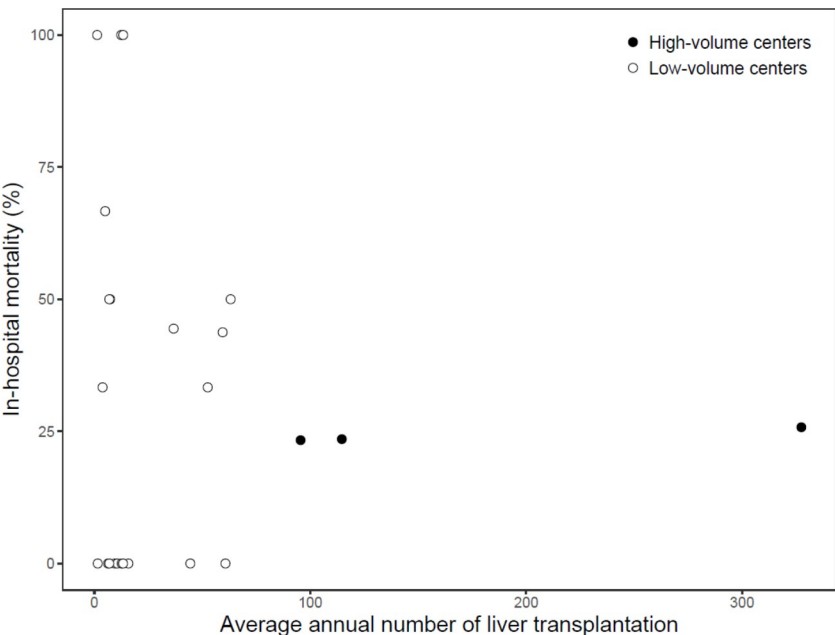

**Fig 2. Relation between the institutional case volume and in-hospital mortality after liver re-transplantation.**

showed a higher mortality rate in low-volume centers (adjusted HR 2.12, 95% CI [1.37, 3.27], $P < 0.001$) compared to high-volume centers (Table 4). The discrepancy in survival rates between high and low-volume centers gradually increased to about 10% by 1 year after liver re-transplantation and was maintained thereafter.

ICU length of stay was similar between high and low-volume centers (26.9 ± 34.5 days vs. 20.2 ± 16.8, $P = 0.094$). However, the hospital length of stay was longer (92.2 ± 76.2 days vs. 67.8 ± 51.5 days, $P = 0.001$) in high-volume centers compared to low-volume centers (Table 5).

## Discussion

The impact of institutional case volume may differ depending on the complexity of the surgical procedure. The case volume effect on postoperative clinical outcome have been reported in complex surgical procedures such as pancreaticoduodenectomy [14] and esophagectomy [15, 16], while institutional case volume did not influence clinical outcomes in relatively simple surgical procedures such as laparoscopic cholecystectomy [23, 24]. Subsequently, complex surgical procedures have been considered to be more suited in high-volume centers with regards to patient outcome.

Previous studies have demonstrated the positive case volume effect in LT. Improved long-term survival and decreased cost in LT patients have been shown using a nationwide cohort database [25] and a cut-off of 20 LTs per year have been suggested to be associated with significantly lower mortality [26, 27]. Recently, we reported that high volume centers had a lower mortality regarding living donor LT and pediatric LT using a nationwide database in Korea [28, 29]. This effect can be attributed to medical resources of higher quality and skill which can affect postoperative outcomes such as experienced personnel including surgeons, anesthesiologists, intensivists, radiologic interventionists, nurses, and pharmacists.

While some studies argue that liver re-transplantation is technically similar to primary LT [30], the surgical procedure involved in liver re-transplantation is considered to be more

**Table 2. Multivariable analysis for in-hospital mortality after liver re-transplantation.**

| | In-hospital mortality (%) | Unadjusted | | Adjusted | |
|---|---|---|---|---|---|
| | | Odds ratio (95% CI) | *P*-value | Odds ratio (95% CI) | *P*-value |
| Age | | | | | |
| 19–50 | 24.5 | Reference | | Reference | |
| 50–60 | 30.2 | 1.33 (0.72, 2.48) | 0.363 | 1.98 (0.95, 4.13) | 0.069 |
| ≥ 60 | 33.3 | 1.54 (0.74, 3.20) | 0.245 | 2.39 (1.02, 5.60) | 0.044* |
| Sex | | | | | |
| Male | 24.6 | Reference | | Reference | |
| Female | 36.3 | 1.75 (1.01, 3.04) | 0.048 | 2.05 (1.07, 3.95) | 0.032* |
| Institutional case volume | | | | | |
| High-volume center (≥ 64 LTs/year) | 25.1 | Reference | | Reference | |
| Low-volume center (< 64 LTs/year) | 36.1 | 1.69 (0.96, 2.96) | 0.069 | 1.93 (1.00, 3.73) | 0.049* |
| Coexisting liver disease | | | | | |
| Hepatitis A virus | 54.5 | 3.16 (0.93, 10.69) | 0.065 | 4.15 (1.09, 15.84) | 0.037* |
| Hepatitis B virus | 26.3 | 0.73 (0.42, 1.27) | 0.271 | 0.99 (0.50, 1.96) | 0.984 |
| Hepatitis C virus | 35.4 | 1.47 (0.76, 2.86) | 0.255 | 1.55 (0.72, 3.34) | 0.267 |
| Hepatocellular carcinoma | 28.0 | 0.95 (0.55, 1.64) | 0.848 | 0.98 (0.50, 1.92) | 0.952 |
| Alcoholic liver cirrhosis | 41.7 | 1.97 (0.96, 4.08) | 0.067 | 1.91 (0.76, 4.77) | 0.167 |
| Primary biliary sclerosis | 28.6 | 0.99 (0.19, 5.24) | 0.995 | 0.81 (0.13, 5.09) | 0.822 |
| Timing of re-transplantation | | | | | |
| Late re-transplantation | 22.8 | Reference | | Reference | |
| Early re-transplantation | 38.5 | 2.12 (1.22, 3.68) | | 1.26 (0.65, 2.45) | 0.049* |
| Donor type of re-transplantation | | | | | |
| Living donor | 6.1 | Reference | | Reference | |
| Deceased donor | 34.0 | 7.89 (2.37, 26.25) | 0.001 | 7.75 (2.13, 28.23) | 0.002 |
| Liver re-transplantation period | | | | | |
| 2014–2016 | 25.3 | Reference | | Reference | |
| 2011–2013 | 31.1 | 1.34 (0.71, 2.53) | 0.371 | 2.11 (1.03, 4.35) | 0.042* |
| 2007–2010 | 30.4 | 1.30 (0.65, 2.57) | 0.459 | 2.07 (0.91, 4.70) | 0.082 |
| Elixhauser comorbidity index | | 1.03 (1.00, 1.07) | 0.058 | 1.03 (0.99, 1.07) | 0.110 |

LT, liver transplantation.

challenging due to a number of reasons including fragile tissue affected by immunosuppression after primary LT and dense vasculature that increases the chance of profuse bleeding during recipient hepatectomy [1, 22]. Therefore, liver re-transplantation is can be considered as one of the most challenging surgical procedures and thus, a difference in outcome according to institutional case volume may be expected. However, studies on the impact of institutional case volume on outcomes after in liver re-transplantation are relatively scarce. Moreover, previous studies were performed in western countries with distinctly different healthcare systems. As the robustness of the case volume effect in liver re-transplantation may become more robust when the relationship can be exhibited under different circumstances, our study serves that purpose as the first Asian study that evaluated the case volume effect in liver re-transplantation. Despite the relatively small number of patients, the implications of our study may lie in that it can serve as a reference to liver transplantations that occur in the Asian region.

Our study results suggests that higher institutional case volume is associated with improved long-term survival after liver re-transplantation. Despite failure to reach statistical significance, high-volume centers showed nearly 10% lower in-hospital mortality compare to low-volume

**Table 3. Multivariable analysis for 1-year mortality after liver re-transplantation.**

| | 1-year mortality (%) | Unadjusted | | Adjusted | |
| --- | --- | --- | --- | --- | --- |
| | | Odds ratio (95% CI) | *P*-value | Odds ratio (95% CI) | *P*-value |
| Age | | | | | |
| 19–50 | 31.9 | Reference | | Reference | |
| 50–60 | 37.2 | 1.27 (0.68, 2.36) | 0.455 | 2.02 (0.94, 4.34) | 0.072 |
| ≥ 60 | 46.5 | 1.86 (0.88, 3.91) | 0.102 | 3.20 (1.29, 7.95) | 0.012* |
| Sex | | | | | |
| Male | 33.1 | Reference | | Reference | |
| Female | 44.4 | 1.62 (0.91, 2.88) | 0.103 | 1.89 (0.93, 3.82) | 0.078 |
| Institutional case volume | | | | | |
| High-volume center (≥ 64 LTs/year) | 32.2 | Reference | | Reference | |
| Low-volume center (< 64 LTs/year) | 46.5 | 1.83 (1.02, 3.26) | 0.041 | 2.54 (1.24, 5.21) | 0.011* |
| Coexisting liver disease | | | | | |
| Hepatitis A virus | 63.6 | 3.19 (0.91, 11.26) | 0.071 | 3.20 (0.79, 13.08) | 0.105 |
| Hepatitis B virus | 34.7 | 0.77 (0.44, 1.37) | 0.375 | 1.19 (0.57, 2.48) | 0.643 |
| Hepatitis C virus | 51.1 | 2.11 (1.09, 4.10) | 0.028 | 2.77 (1.25, 6.11) | 0.012* |
| Hepatocellular carcinoma | 33.3 | 0.77 (0.44, 1.35) | 0.246 | 0.58 (0.29, 1.17) | 0.129 |
| Alcoholic liver cirrhosis | 38.7 | 1.10 (0.50, 2.40) | 0.814 | 1.10 (0.40, 3.03) | 0.849 |
| Primary biliary sclerosis | 50.0 | 1.74 (0.34, 8.85) | 0.502 | 2.13 (0.33, 13.90) | 0.430 |
| Timing of re-transplantation | | | | | |
| Late re-transplantation | 30.4 | Reference | | Reference | |
| Early re-transplantation | 47.6 | 2.07 (1.18, 3.65) | 0.012 | 1.59 (0.76, 3.33) | 0.218 |
| Donor type of re-transplantation | | | | | |
| Living donor | 17.0 | Reference | | Reference | |
| Deceased donor | 42.2 | 3.56 (1.57, 8.07) | 0.002 | 3.77 (1.43, 9.95) | 0.007* |
| Liver re-transplantation period | | | | | |
| 2014–2016 | 24.6 | Reference | | Reference | |
| 2011–2013 | 37.8 | 1.86 (0.91, 3.83) | 0.092 | 2.90 (1.27, 6.632) | 0.011* |
| 2007–2010 | 46.4 | 2.65 (1.25, 5.62) | 0.011 | 4.07 (1.65, 10.04) | 0.002* |
| Elixhauser comorbidity index | | 1.03 (1.00, 1.07) | 0.068 | 1.03 (0.99, 1.08) | 0.138 |

LT, liver transplantation.

centers. Similar results were have been reported in a study using US registry data which showed that 1-year patient survival after liver re-transplantation was superior in high-volume centers [31].

The results of our study may be used as a supportive evidence for centralization/regionalization. Patients requiring complex and high risk surgical procedures may anticipate a better outcome when they receive care in designated centers with sufficient experience. Centralization/regionalization was initially emphasized in children, especially in congenital pediatric disease with extremely low incidence [32–34]. Recently, it has gained interest in adults along with the concept of accountable health care systems with acceptable costs [35]. As shown in Fig 1, the mortality rate is relatively constant in the high-volume centers; while in the low-volume centers, there are many centers with such a high mortality rate more than 50%. In addition, despite the higher proportion of living donor LT, which is more challenging than deceased donor LT, the high-volume centers showed lower incidence of early re-transplantation compared to low-volume centers. Since the incidence of early re-transplantation due to primary non-function, small-for-size syndrome, or early hepatic artery thrombosis after LT can be

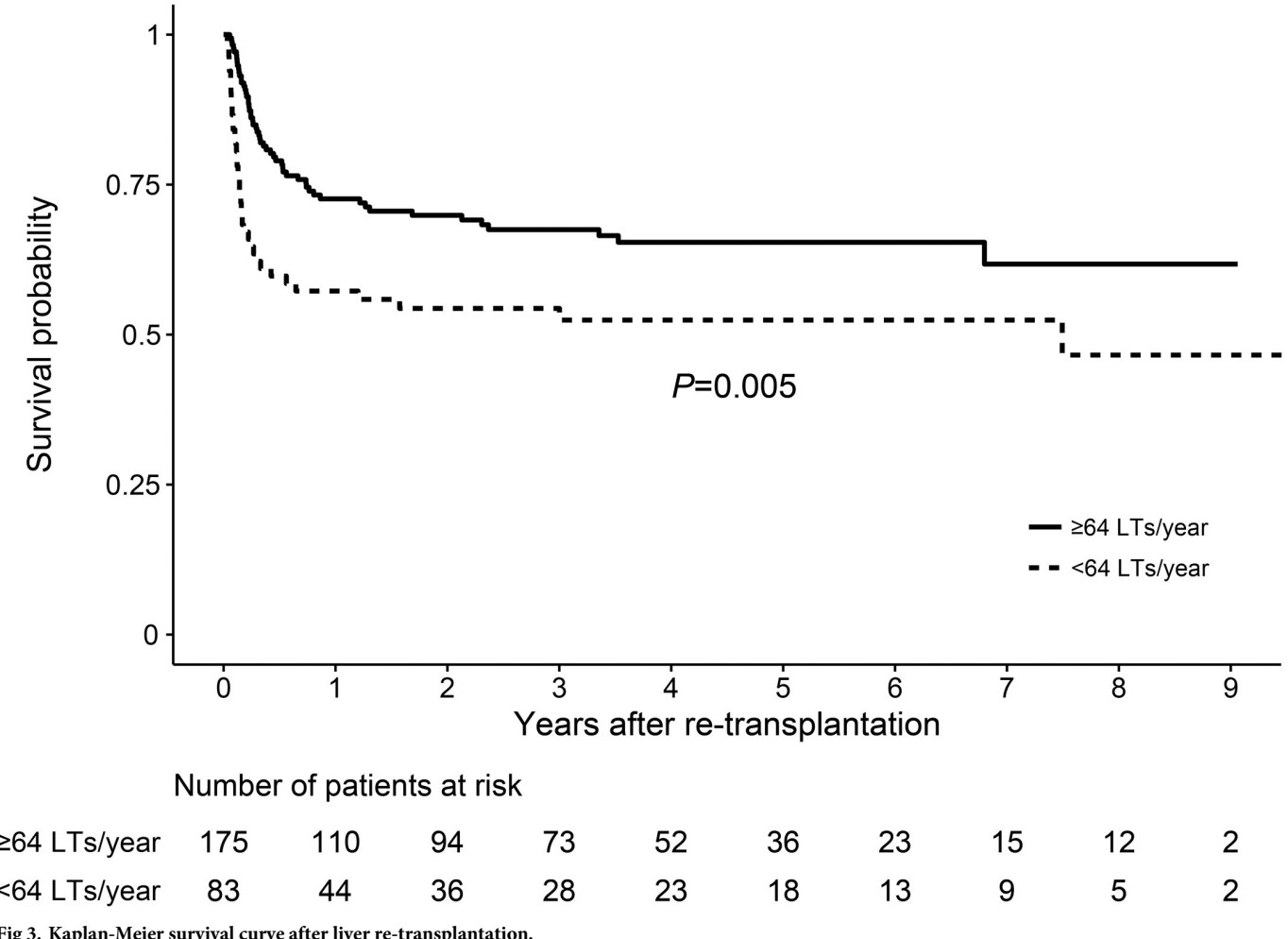

**Fig 3. Kaplan-Meier survival curve after liver re-transplantation.**

considered as a performance indicator, this result may be another piece of evidence supporting centralization of LT. Considering these points, we suggest that there should be a cut-off for minimal case volume in centers performing liver re-transplantation. However, it cannot be generalized that all re-transplantations should be performed in high-volume centers. The cause of early re-transplantation may be technical issues, but it may also be misjudgment. Therefore, early re-transplantation may occur more frequently in inexperienced low-volume centers compared to high-volume centers. Since most of early re-transplantations are performed in critically ill patients with rapidly progressing hepatic failure, transplantations must be performed as quickly as possible, and transferring patients to high-volume centers may be risky. Therefore, centralization may be more suitable for late re-transplantations in which the procedure may be performed in an elective basis.

There is no clear cut-off that separates between high and low-volume centers for evaluating patient outcome after liver re-transplantation. In the study that evaluated patient outcome after liver re-transplantation according to case volume using US registry data, centers were divided into tertiles of approximately equal size and the annual number of LTs performed in high-volume centers ranged from 88 to 210 LTs [31]. The cut-off for dividing high and low-volume centers in our study was 64 LTs per year. Considering the cut-offs of previous studies, it may be inappropriate to classify a center that performs more than 50 LTs per year as a

**Table 4. Multivariable analysis for overall mortality after liver re-transplantation.**

| | Unadjusted | | Adjusted | |
|---|---|---|---|---|
| | Hazard ratio (95% CI) | P-value | Hazard ratio (95% CI) | P-value |
| Age | | | | |
| 19–50 | Reference | | Reference | |
| 50–60 | 1.24 (0.79, 1.95) | 0.348 | 1.44 (0.87, 2.38) | 0.160 |
| ≥ 60 | 1.80 (1.11, 2.94) | 0.018* | 2.33 (1.36, 4.01) | 0.002* |
| Sex | | | | |
| Male | Reference | | Reference | |
| Female | 1.55 (1.05, 2.28) | 0.026* | 1.73 (1.13, 2.65) | 0.012* |
| Institutional case volume | | | | |
| High-volume center (≥ 64 LTs/year) | Reference | | Reference | |
| Low-volume center (< 64 LTs/year) | 1.85 (1.26, 2.72) | 0.002* | 2.12 (1.37, 3.27) | 0.001* |
| Coexisting liver disease | | | | |
| Hepatitis A virus | 2.33 (1.18, 4.62) | 0.015* | 2.81 (1.30, 6.05) | 0.008* |
| Hepatitis B virus | 0.86 (0.58, 1.27) | 0.450 | 1.08 (0.70, 1.67) | 0.740 |
| Hepatitis C virus | 1.55 (1.01, 2.37) | 0.046 | 1.78 (1.11, 2.85) | 0.017* |
| Hepatocellular carcinoma | 1.11 (0.76, 1.62) | 0.605 | 1.07 (0.68, 1.68) | 0.772 |
| Alcoholic liver cirrhosis | 1.10 (0.65, 1.88) | 0.719 | 1.08 (0.58, 2.00) | 0.813 |
| Primary biliary sclerosis | 1.12 (0.35, 3.52) | 0.851 | 1.07 (0.31, 3.71) | 0.911 |
| Timing of re-transplantation | | | | |
| Late re-transplantation | Reference | | Reference | |
| Early re-transplantation | 1.68 (1.15, 2.46) | 0.008* | 1.24 (0.79, 1.95) | 0.350 |
| Donor type of re-transplantation | | | | |
| Living donor | Reference | | Reference | |
| Deceased donor | 2.82 (1.51, 5.29) | 0.001* | 2.55 (1.31, 4.98) | 0.006* |
| Elixhauser comorbidity index | 1.03 (1.01, 1.05) | 0.031* | | |

LT, liver transplantation.

low-volume center. However, with no previous studies that may serve as a reference, the cut-off was set at 64 LTs per year based on the scatterplot of annual number of LTs per center, disproportionate distribution of LTs in Korea, and the ROC curve analysis which determined the optimal cut-off value at 63.21 LTs per year.

**Table 5. Outcomes after liver re-transplantation.**

| | Low-volume center (< 64 LTs/year) | High-volume center (≥ 64 LTs/year) | P-value |
|---|---|---|---|
| In-hospital mortality | 36.1% | 25.1% | 0.069 |
| ICU length of stay (days) | 20.2 ± 16.8 | 26.9 ±34.5 | 0.094 |
| ICU length of stay in survivors (days) | 18.2 ± 15.5 | 17.4 ± 22.6 | 0.818 |
| Hospital length of stay (days) | 67.8 ± 51.5 | 92.2 ± 76.2 | 0.009* |
| Hospital length of stay in survivors (days) | 72.2 ± 52.5 | 86.7 ± 71.2 | 0.183 |

Data are presented as % or mean ± SD.

ICU, intensive care unit.

The interval between primary LT and re-transplantation has been suggested to impact clinical outcomes after liver re-transplantation. Early liver re-transplantation is technically less challenging compared to late liver re-transplantation because there is less adhesion and newly developed portal venous collaterals make late liver re-transplantation difficult [36]. The similar in-hospital mortality after liver re-transplantation between the high and low-volume centers in our study may partly be explained by the higher proportion of early liver re-transplantation in the low-volume centers compared to the high volume centers. The difference in outcome between early and late liver re-transplantation are still controversial as some studies suggest that early re-transplantation is associated with better survival [37, 38], whereas others report that late re-transplantation may result in better outcome [17].

The cause of graft failure is another important factor associated with clinical outcome after liver re-transplantation. Primary non-function, hepatic artery thrombosis, acute or chronic rejection, disease recurrence, and biliary complications account for more than 90% of liver re-transplantation. Since primary non-function and hepatic artery thrombosis occur relatively early after primary LT compared to other causes [5, 12], the cause of graft failure can be presumed through the interval between the primary LT and re-transplantation, or vice versa. Considering the pattern of LT in Korea where the majority of primary LTs is living donor LT [29], early re-transplantation due to hepatic artery thrombosis may be more likely to occur in a few high-volume centers where most living donor LTs are concentrated whereas low-volume centers may be more likely to perform deceased donor LT for both primary LT and liver re-transplantation. Among the other causes of graft failure, recurrent HCV and primary non-function after primary LT are risk factors associated with worse outcome after liver re-transplantation [5, 39].

Higher Model for end-stage liver disease (MELD) scores and older recipient's age are also established risk factors for postoperative mortality after liver re-transplantation. MELD scores, which are used to prioritize liver transplant candidates in many countries including Korea, have been shown to correlate with postoperative survival after liver re-transplantation with an estimated 50% mortality in patients with MELD scores over 30 [11, 40]. The lack of MELD score data is one of the limitations of our study which was inherent to the administrative nature of the database. Interestingly, female sex was a newly identified risk factor of mortality after liver re-transplantation. Although the underlying mechanism is unclear, our results suggest that female patients may be at a greater risk of both in-hospital and overall mortality after liver re-transplantation.

Contrary to previous reports [25], the hospital length of stay was longer in high-volume centers compared to low-volume centers. Considering that the ICU length of stay was similar, a potential underlying cause may be the higher in-hospital mortality in low-volume centers. Another explanation may be the reimbursement scheme of the Korean healthcare system which is predominantly pay-for service and thus, institutions are not penalized for prolonged hospital length of stay. Tendency of earlier decision to re-transplant in high-volume centers may also have contributed.

The completeness of the NIHS database in coverage of the whole Korean population may be another strength of our study. The healthcare system in Korea is a single payer system and insures more than 97% of residents in Korea [41] with equal benefits to all, regardless of insurance premiums that differ depending on income. The remaining 3% of the population with lowest income are supported by the Medical Aid program. The bulk of the incurred medical expense is reimbursed by the NHIS and the details of the claim is stored in the NHIS database. Although the relatively small number of liver re-transplantation patients who are not eligible for NHIS coverage are not included in the study, the database used for analysis includes all remaining patients who received liver transplantation and re-transplantation in Korea with

complete follow-up using the resident registration number, a personal identifier assigned to every Korean citizen. Therefore, our study is free from *selection bias*, or incomplete/missing data regarding outcomes [1, 10, 31]. Due to the completeness of the NHIS database, we believe that our results reflect the real world and the most recent outcomes.

Our study has several limitations to consider. First, the database used in this study was not intended for clinical research and many clinically relevant information was not available and therefore, not analyzed. Reported factors associated with survival after liver re-transplantations such as MELD scores and cause of graft failure after primary LT were not available. Currently, there are no nationwide database which contains all clinical data for every patient undergoing liver re-transplantation. As in other recent studies that used administrative data, we attempted to best adjust the severity of illness by using the Elixhauser comorbidity index in addition to the basic baseline characteristics. Second, due to the complexity of surgical procedure and postoperative management, there may be substantial heterogeneity in real practices depending on institutions. Third, since only 3 centers were classified as high-volume centers, there is a potential for skewed results. However, the 3 centers performed more than two thirds of all liver re-transplantations. Therefore, a statistical approach of dividing centers (ex. quartiles or tertiles) may introduce other biases due to the skewed distribution of liver re-transplantation cases.

## Conclusion

The 1-year mortality after liver re-transplantation appeared to be significantly lower in centers with higher case volume centers more than 64 LTs per year compared to lower case volume centers less than 64 LTs per year. The positive effect of institutional case volume suggests that there may be an opportunity for quality improvement in liver re-transplantation.

## Author Contributions

**Conceptualization:** Seung-Young Oh, Hannah Lee, Ho Geol Ryu.

**Data curation:** Eun Jin Jang, Ga Hee Kim.

**Formal analysis:** Eun Jin Jang, Ga Hee Kim, Seokha Yoo.

**Investigation:** Seung-Young Oh, Ga Hee Kim, Hannah Lee, Ho Geol Ryu.

**Methodology:** Seung-Young Oh, Bo Rim Kim, Ho Geol Ryu.

**Supervision:** Ho Geol Ryu.

**Writing – original draft:** Seung-Young Oh, Eun Jin Jang, Seokha Yoo.

**Writing – review & editing:** Seung-Young Oh, Eun Jin Jang, Hannah Lee, Nam-Joon Yi, Bo Rim Kim, Ho Geol Ryu.

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
