## [Decision Letter · Decision Letter 0]

10 Nov 2020

PONE-D-20-31343

Association between hospital liver transplantation volume and mortality after liver re-transplantation

PLOS ONE

Dear Dr. Ryu,

Thank you for submitting your manuscript to PLOS ONE. After careful consideration, we feel that it has merit but does not fully meet PLOS ONE’s publication criteria as it currently stands. Therefore, we invite you to submit a revised version of the manuscript that addresses the points raised during the review process.

The MS by Ryu and colleagues is certainly of interest, and has been reviewed by two expert reviewers. They have come back with the request for revisions, esp relating to analyses and interpretation/discussion of some of the data. Hopefully authors can make the revisions required and respond to the reviewers in a point-by-point fashion.

We look forward to receiving your revised manuscript.

Kind regards,

Frank JMF Dor, M.D., Ph.D., FEBS, FRCS

Academic Editor

PLOS ONE

Journal Requirements:

2. Please include the date(s) on which you accessed the databases or records to obtain the data used in your study.

Reviewers' comments:

Reviewer's Responses to Questions

**Comments to the Author**

1. Is the manuscript technically sound, and do the data support the conclusions?

Reviewer #1: Yes

Reviewer #2: Yes

2. Has the statistical analysis been performed appropriately and rigorously? 

Reviewer #1: Yes

Reviewer #2: Yes

3. Have the authors made all data underlying the findings in their manuscript fully available?

Reviewer #1: No

Reviewer #2: Yes

4. Is the manuscript presented in an intelligible fashion and written in standard English?

Reviewer #1: Yes

Reviewer #2: Yes

5. Review Comments to the Author

Reviewer #1: This retrospective study analyzed national insurance data regarding liver re-transplantation in the Republic of Korea (a.k.a. South Korea). It is a well written and interesting manuscript, however some points need clarification.

1. Was there a significant difference in the incidence of re-transplantation between high-volume and low-volume centers (defined as performing <64 or >= 64 LT annually) ? Background: the incidence of re-transplantation may also be seen as a performance indicator of a center especially with regard to incidence of primary non-function (PNF) , small-for-size-syndrome (SFSS), or early Hepatic artery thrombosis (HAT).

2. Was there any difference between high-volume and low-volume centers regarding donor source (i.e. DBD vs. DCD vs. LDLT ?)

3. the age grouping is confusing, it would be better to replace it with median recipient age and range.

4. The relationship between timing of re-transplantation and outcome needs a discussion that is even more nuanced and differentiating. The AU themselves point out that early re-transplantation is often technically easier, however graft quality may be compromised (since it is mostly an emergency procedure that has to be carried out a.s.a.p - whereas late re-LT is often an elective procedure), furthermore due to the emergency character patients are often in critical care with multi-organ failure. Now add to this that the need of early re-transplantation may a consequence of errors in judgement and/ or performance (PNF, SFSS, HAT see above) it comes as no surprise that early re-transplants occur more often at low-volume centers and have a worse outcome. Referral of a patient in this scenario from a low-volume to a high-volume center does not seem realistic and probably would not change the outcome. However what may follow from these data would be a recommendation of referral of late re-transplants to high-volume centers.

Reviewer #2: I think this is an interesting and original retrospective study evaluating the impact of centre volume on the outcome of liver re-LT.

The main limit of this is well underlined by the authors in the discussion and it is tha absence of some relevant variables in the analysis (i.e. MELD score, HCC characteristics).

I have only some minor comments.

1) The authors found 64 as cut off to define high and low volume centers using ROC curve. However, it is not clear to me what was the endpoint of the ROC curve ? It was in hospital mortality, or 1-year moralitity, or other?

2) Reading the Methods section it seems that surgical tecnique of first LT and re-LT (LDLT vs. DDLT) probably is available for the authors. Why did not include this variable in logistic and Cox regressions?

6. PLOS authors have the option to publish the peer review history of their article (what does this mean?). If published, this will include your full peer review and any attached files.

Reviewer #1: **Yes: **Prof.Dr.Wolf O. Bechstein, MD

Reviewer #2: **Yes: **UMBERTO CILLO

---

## [Author Response · Author response to Decision Letter 0]

31 May 2021

Dear Editor and Reviewers

First of all, we would like to express our gratitude for the constructive comments by the reviewers that has helped us significantly improve the quality of our manuscript. And we sincerely appreciate the extension of the deadline for submitting revisions even though the waiting time for reanalysis was long due to the priority of COVID-19 related analysis. As we went through the comments and questions, we were grateful for the time and effort the reviewers have obviously made to point out important aspects. A response letter containing the answer to the reviewer's comment is attached as a separate file. 

Sincerly, 

Ho Geol Ryu

Department of Anesthesiology and Pain Medicine, Seoul National University Hospital,

Seoul National University College of Medicine,

101 Daehak-ro, Jongno-gu, Seoul, Korea, 03080

Tel: 82-2-2072-2065, Fax: 82-2-747-5639, E-mail: hogeol@gmail.com

---

## [Decision Letter · Decision Letter 1]

22 Jul 2021

Association between hospital liver transplantation volume and mortality after liver re-transplantation

PONE-D-20-31343R1

Dear Dr. Ryu,

We’re pleased to inform you that your manuscript has been judged scientifically suitable for publication and will be formally accepted for publication once it meets all outstanding technical requirements.

Kind regards,

Frank JMF Dor, M.D., Ph.D., FEBS, FRCS

Academic Editor

PLOS ONE

Additional Editor Comments (optional):

Reviewers' comments:

Reviewer's Responses to Questions

**Comments to the Author**

1. If the authors have adequately addressed your comments raised in a previous round of review and you feel that this manuscript is now acceptable for publication, you may indicate that here to bypass the “Comments to the Author” section, enter your conflict of interest statement in the “Confidential to Editor” section, and submit your "Accept" recommendation.

Reviewer #2: All comments have been addressed

2. Is the manuscript technically sound, and do the data support the conclusions?

Reviewer #2: Yes

3. Has the statistical analysis been performed appropriately and rigorously? 

Reviewer #2: Yes

4. Have the authors made all data underlying the findings in their manuscript fully available?

Reviewer #2: Yes

5. Is the manuscript presented in an intelligible fashion and written in standard English?

Reviewer #2: Yes

6. Review Comments to the Author

Reviewer #2: I think that the authors have adequately addressed my comments raised in a previous round of review and I feel that this manuscript is now acceptable for publication.

7. PLOS authors have the option to publish the peer review history of their article (what does this mean?). If published, this will include your full peer review and any attached files.

Reviewer #2: No

---

## [Editor Report · Acceptance letter]

27 Jul 2021

PONE-D-20-31343R1 

Association between hospital liver transplantation volume and mortality after liver re-transplantation 

Dear Dr. Ryu:

I'm pleased to inform you that your manuscript has been deemed suitable for publication in PLOS ONE. Congratulations! Your manuscript is now with our production department. 

Kind regards, 

on behalf of

Dr. Frank JMF Dor 

Academic Editor

PLOS ONE